# Suramin Affects the Renal VEGF-A/VEGFR Axis in Short-Term Streptozotocin-Induced Diabetes

**DOI:** 10.3390/ph16030470

**Published:** 2023-03-22

**Authors:** Gabriela Chyła-Danił, Kornelia Sałaga-Zaleska, Ewelina Kreft, Aleksandra Krzesińska, Sylwia Herman, Agnieszka Kuchta, Monika Sakowicz-Burkiewicz, Małgorzata Lenartowicz, Maciej Jankowski

**Affiliations:** 1Department of Clinical Chemistry, Medical University of Gdańsk, Dębinki 7, 80-210 Gdańsk, Poland; 2Laboratory of Genetics and Evolutionism, Institute of Zoology and Biomedical Research, Jagiellonian University, Gronostajowa 9, 30-387 Kraków, Poland; 3Department of Molecular Medicine, Medical University of Gdańsk, Dębinki 7, 80-210 Gdańsk, Poland

**Keywords:** VEGF-A, vascular endothelial growth factor A, VEGFR1, VEGFR2, diabetes mellitus, diabetic nephropathy, suramin, streptozotocin

## Abstract

Diabetic nephropathy (DN) accounts for approximately 50% of end-stage renal diseases. Vascular endothelial growth factor A (VEGF-A) is thought to be a critical mediator of vascular dysfunction in DN, but its role is unclear. The lack of pharmacological tools to modify renal concentrations further hinders the understanding of its role in DN. In this study, rats were evaluated after 3 weeks of streptozotocin-induced diabetes and two suramin treatments (10 mg/kg, *ip*). Vascular endothelial growth factor A expression was evaluated by western blot of glomeruli and immunofluorescence of the renal cortex. RT-PCR for receptors *Vegfr1* mRNA and *Vegfr2* mRNA quantitation was performed. The soluble adhesive molecules (sICAM-1, sVCAM-1) in blood were measured by ELISA and the vasoreactivity of interlobar arteries to acetylcholine was evaluated using wire myography. Suramin administration reduced the expression and intraglomerular localisation of VEGF-A. Increased VEGFR-2 expression in diabetes was reduced by suramin to non-diabetic levels. Diabetes reduced the sVCAM-1 concentrations. Suramin in diabetes restored acetylcholine relaxation properties to non-diabetic levels. In conclusion, suramin affects the renal VEGF-A/VEGF receptors axis and has a beneficial impact on endothelium-dependent relaxation of renal arteries. Thus, suramin may be used as a pharmacological agent to investigate the potential role of VEGF-A in the pathogenesis of renal vascular complications in short-term diabetes.

## 1. Introduction

Diabetes mellitus (DM) is a group of metabolic disorders characterised by hyperglycaemia and is becoming a global health burden, with a 10.5% prevalence worldwide, as estimated in 2021 [1]. Hyperglycaemia stimulates the synthesis and secretion of growth factors, including vascular endothelial growth factors (VEGF), and triggers a number of interrelated metabolic and haemodynamic effects that contribute to an increase in growth factors and lead to diabetic microvascular complications, including diabetic nephropathy (DN) [2]. Diabetic nephropathy is morphologically manifested by glomerular hypertrophy, podocyte and endothelial cell damage [3]. These pathological abnormalities result in the decline of the glomerular filtration rate (GFR) and the increase of glomerular filter permeability to albumin, resulting in increased albumin excretion in the urine [4]. Affecting up to 40% of patients with diabetes, DN may lead to end-stage renal disease, which is one of the leading causes of premature death [4]. Importantly, decreased endothelial cell survival and systemic vascular endothelial dysfunction, affecting vascular tone adjustment and permeability status, are the initiating stages of microvascular complications in diabetes [5,6]. The dysfunction of endothelial cells promotes the progression of DN [7].

Vascular endothelial growth factor A (VEGF-A) is a major member of the VEGF family of cytokines, along with VEGF-B, -C, -D and -E and placental growth factor (PlGF). The *VEGF-A* gene can be alternatively spliced to encode several isoforms that have unique biological functions, and the biological relevance of this is being investigated [8]. Vascular endothelial growth factor A is a potent angiogenic paracrine factor, whose activities include endothelial cell survival, proliferation, migration and tube formation [9]. In addition, autocrine VEGF-A signalling is required for vascular homeostasis [10]. It has also been shown to be required for the maintenance of the differentiated phenotype in endothelial cells, for example, the presence of endothelial fenestrations in glomeruli [11]. Vascular endothelial growth factor A also acts as a proinflammatory cytokine by increasing endothelial permeability and inducing adhesion molecules, including the intercellular adhesion molecule 1 (ICAM-1) and vascular cell adhesion molecule 1 (VCAM-1), which bind leukocytes to endothelial cells [12]. The distinct signal transduction mechanisms by which VEGF-A induces survival, proliferation, migration and nitric oxide (NO) production in endothelial cells are still under investigation [13]. Cellular responses to VEGF-A are primarily driven by its binding to cell surface receptors tyrosine kinases, known as VEGFR-1 and VEGFR-2. Most notably, it interacts with VEGFR-2 [14,15]. In contrast, VEGFR-1 seems to have an inhibitory role by sequestering VEGF-A, thereby preventing its interaction with VEGFR-2 [16]. Both VEGF-A and its receptors are expressed in the kidneys. Electron microscopy studies have shown that VEGF-A is localised in the foot processes of podocytes, which are the main sources of VEGF-A in the kidney, glomerular basement membrane, the lumen and on the endothelial surface, indicating that VEGF-A can be transported into endothelial cells in a direction opposite to the glomerular filtrate flow [17]. Additionally, receptors for VEGF-A are predominantly expressed on the endothelium of glomeruli, pre- and postglomerular arteries and, to a lesser extent, on mesangial cells and tubular cells [18]. The expression of VEGF-A and related receptors in the glomerulus and their biological roles suggest their potential involvement in DN [19]. Studies in streptozotocin-induced diabetic rats have depicted the upregulation of VEGF-A and VEGFR-2 in the kidneys during the early stages of DN, and glomeruli have been identified as the main site of VEGF-A binding [20]. Furthermore, it has been reported that in DN, there is an increase in plasma VEGF levels in both adults and children with diabetes [21].

Although VEGF-A was initially ascribed a negative role in the development of DN, many current reports indicate that the initial view of the role of VEGF-A was very one-sided, and the complex action of this cytokine has now been postulated [22]. This is supported by the results of studies in which the glomerular-specific depletion or overexpression of VEGF-A mice leads to glomerular pathology, suggesting the importance of a balance in VEGF-A expression [23]. Furthermore, the deletion of all VEGF-A isoforms from podocytes accelerates nephropathy in diabetic animals [24], and podocyte VEGF-A knockdown induces diffuse glomerulosclerosis in diabetic kidneys [25]. Recently, it was shown that isoform VEGF-A_165b_ via VEGFR-2 normalises glomerular permeability to albumin [26]. In humans, molecular analysis of microdissected tubulointerstitial compartments from biopsies has provided evidence that VEGF-A expression is decreased in patients with diabetes [27]. Moreover, it has been suggested that the local interaction between VEGF-A and its receptors, but not VEGF-A expression, is tightly regulated in diabetes [28] The ambiguous role of VEGF-A in the development of DN is also reflected in the results of therapies based on the inhibition of VEGF-A-induced signalling, leading to increased protein excretion in the urine, among other things, suggesting that certain amounts of VEGF-A are needed to maintain normal glomerular function [29]. Thus, it is important to find a pharmacological or biochemical way to modulate the renal expression of VEGF-A in diabetes.

Our previous studies have shown that suramin increases the excretion of VEGF-A in the urine in short-term diabetes caused by streptozotocin, while at the same time, there are no significant changes in the blood concentration of this cytokinin. This may indicate that changes in the expression of VEGF-A in the kidneys occur under the influence of suramin [30]. Suramin is used in the laboratory as a broad-spectrum antagonist of P2Rs and also clinically with a wide array of potential applications in parasite-induced diseases, oncology, psychiatry and inflammatory diseases of the cardiovascular system [31]. Its action leads to reduced inflammation and fibrosis, including advanced fibrosis and improved renal function, such as reduced proteinuria in type 2 diabetes patients [32,33,34]. The mechanism of the effect of suramin is not completely known. However, it is known to interfere with the action of certain growth factors by competitive binding to their receptors [35]. Suramin mostly accumulates in the kidneys and has a long half-life, which requires it to be administered weekly [36]. As has been previously shown, suramin administered intraperitoneally at 10 mg/kg once per week for two weeks prevents 24-h proteinuria, attenuated renal fibrosis and glomerular damage from arising in the remnant kidney model of chronic kidney disease. Suramin also reduces the expression of mothers against decapentaplegic (SMAD) homolog 3, epidermal growth factor receptor, platelet-derived growth factor receptor, nuclear factor-κB, extracellular-regulated kinase 1 and signal transducer and activator of transcription. It reduces the recruitment of monocytes and macrophages to the kidney by decreasing the expression of monocyte chemoattractant protein-1 [32].

The aim of our study was to assess the potential role of suramin in the regulation of renal VEGF-A and its receptors and in regulating renal endothelial function in a research model of short-term streptozotocin-induced diabetes. To achieve this objective, renal VEGF-A protein levels, *Vegfr* mRNA expression levels and the endothelial function of renal interlobar arteries were determined in short-term (3 weeks) streptozotocin-induced diabetic rats treated with suramin. Our results suggest that suramin may be considered a pharmacological tool for affecting the VEGF-A/VEGFR axis.

## 2. Results

### 2.1. Experiments In Vivo

The data on body weight, fluid intake, urine excretion and blood glucose concentration in the rats in all the experimental groups are presented in Figure 1. Suramin-untreated non-diabetic rats (CON) and suramin-treated non-diabetic rats (SUR) showed a significant increase in body weight (17%) over 2 weeks. No significant changes in body weight were observed in the diabetic rats not treated with suramin (STZ) or those treated with suramin (STZ+SUR). The nutrition variables, i.e., food intake, specific rate of body mass gain, feed efficiency ratio (FER) and efficiency of food utilisation for body weight (EFU_BW_) are shown in Table 1. Accordingly, diabetic rats had a 1.5-fold higher food intake and a slight but significantly lower specific rate of body mass gain FER and EFU_BW_ compared to non-diabetic rats. Suramin did not significantly affect these nutritional variables in either diabetic or non-diabetic rats. The diabetic rats had higher diuresis (12-fold) and water intake (5-fold) compared to the non-diabetic rats (Figure 1). Suramin did not significantly affect these parameters in either diabetic or non-diabetic rats.

### 2.2. Experiments In Vitro

To check the potential influence of suramin on the glomerular filtration process, we estimated the glomerular filtration rate. Figure 2A–C show the results of creatinine (A) and urea (B) concentrations in serum and the estimated glomerular filtration rate (C). There were no differences in creatinine concentrations among the study groups (Figure 2A). However, the urea concentration was 11% higher in the diabetic rats compared to the control rats (8.4 ± 0.8 mmol/L vs. 7.6 ± 0.4 mmol/L, F_1,15_ = 4.801, *p* = 0.0446), and the two-way ANOVA did not indicate an impact of suramin on the demonstrated differences (F_1,15_ = 0.8335, *p* = 0.3757) (Figure 2B). Concentrations of creatinine and urea were used to estimate glomerular filtration rates; there was an average of 13% lower eGFR in the diabetic group compared to the control rats (2078 ± 108 µL/min vs. 2377 ± 123 µL/min, F_1,15_ = 9.366, *p* = 0.0079), and suramin did not impact the observed differences (F_1,15_ = 1.917, *p* = 0.1865) (Figure 2C).

The data regarding the VEGF-A protein level in the isolated glomeruli are shown in Figure 3. Diabetic glomeruli were characterised by a 54% reduction of VEGF-A protein level in glomeruli isolated compared to the control group (0.18 ± 0.02 vs. 0.33 ± 0.03, *F*_1,22_ = 11.86, *p* = 0.0023). Our results also show that suramin had no significant effect on VEGF-A levels in glomeruli isolated either from non-diabetic (*p* = 0.9513) or diabetic rats (*p* = 0.9824) (Figure 3A,B).

To further explore the effect of suramin on VEGF-A expression in the kidneys, an immunofluorescence study was performed. The expression and localisation of the VEGF-A protein in the kidneys of non-diabetic and diabetic rats treated or untreated with suramin are shown in Figure 4. Fluorescence staining of kidney scrapings from all the groups of the rats tested using anti-VEGF-A antibody showed positive signals in the glomerular cells (Figure 4A–D). Diabetes did not significantly affect VEGF-A expression in the glomeruli (Figure 4B,F). However, suramin significantly reduced VEGF-A expression in glomeruli by 40% in both non-diabetic and diabetic rats (7.19 ± 0.37 vs. 11.81 ± 0.84, *F*_1, 28_ = 20.83, *p* < 0.0001, Figure 4F). Interestingly, suramin also altered the localisation of VEGF-A expression in non-diabetic (Figure 4C) and diabetic (Figure 4D) rats; namely, the positive signal mainly corresponded to the localisation of parietal epithelial cells.

The expression and localisation of VEGF-A protein in the kidneys of non-diabetic and diabetic rats treated with suramin is presented in Figure 5. Fluorescence immunostaining of kidney sections from all the study groups of the rats using an anti-VEGF-A antibody showed positive signals in cells of the descending limb of the loop of Henle.

Next, the expression of receptors for VEGF-A (*Vegfr1* and *Vegfr2*) mRNAs in the renal cortex was measured quantitatively by RT-PCR (Figure 6). There was no difference in the *Vegfr1* mRNA levels between the diabetic and non-diabetic rats (*F*_1,19_ = 1.387, *p* = 0.2535). We also observed no effect of suramin on *Vegfr1* mRNA levels in both groups (*F*_1,19_ = 0.015, *p* = 0.9036) (Figure 6A). However, we observed different effects of suramin on the *Vegfr2* mRNA levels in the non-diabetic and diabetic rats (*F*_1,19_ = 15.08, *p* = 0.001) (Figure 6B). In the diabetic rats, suramin reduced the *Vegfr2* mRNA levels by an average of 84% (*p* = 0.0001). In contrast, there was no effect of suramin in the non-diabetic rats (*p* = 0.9787). At the same time, the observed difference between diabetic and non-diabetic rats was only significant in rats not treated with suramin (*p* = 0.0002).

Since VEGF-A may affect the expression of the vascular cell adhesion molecule-1 (VCAM-1) and intercellular adhesion molecule-1 (ICAM-1) susceptible to proteolytic cleavage, resulting in the production of soluble forms, the next experiments were performed to measure the serum concentrations of the soluble forms of VCAM-1 (sVCAM-1) and ICAM-1 (sICAM-1). The suramin did not significantly influence the sVCAM-1 concentration in the diabetic and non-diabetic rats (*F*_1,23_ = 2.063, *p* = 0.1643) (Figure 7A). There was a significant reduction (about 52%) in the sVCAM-1 concentrations in diabetic rats compared to non-diabetic rats in both animals that were treated and not treated with suramin (*F*_1,23_ = 21.21, *p* = 0.0001). Analysis of the sICAM-1 levels showed no differences between the diabetic and non-diabetic rats (*F*_1,24_ = 1.769 × 10^−5^, *p* = 0.9967) (Figure 7B). There was also no effect of suramin on sICAM-1 concentrations (*F*_1,24_ = 0.7933, *p* = 0.3819).

### 2.3. Experiments Ex Vivo

Wire myography of renal interlobar arteries (ILA) was used to study the effect of acetylcholine (10^–9^–10^–5^ M) on phenylephrine-precontracted ILA isolated from non-diabetic and diabetic rats treated or untreated with suramin (Figure 8). The concentration-dependent effects of acetylcholine were observed for all the experimental groups (Figure 8A), and the maximal relaxation values are presented in Figure 8B. In the control group, a minimal relaxing effect of about 40% (*p* = 0.0017) occurred at 10^–7^ M acetylcholine, but a maximal effect of about 50% (*p* < 0.001) was observed at a concentration of 10^–6^ M. This further increased the concentration of acetylcholine up to 10^–5^ M and did not significantly enhance its relaxing effects. Suramin administration to non-diabetic rats shifted the value of the minimal effective concentration up to 10^–6^ M with an effect of about 42% (*p* = 0.0236). A maximal relaxing effect of about 45% (*p* = 0.0197) was observed at 5 × 10^–5^ M and was not statistically different from the maximal effect observed in the control rats (Figure 8B). In the diabetic rats, the minimal effective concentration of acetylcholine at 10^–6^ M simultaneously induced a maximal relaxing effect of about 19% (*p* = 0.0065), which was significantly lower compared to the control rats (*p* < 0.05, Figure 8B). Furthermore, suramin administration to diabetic rats induced the maximal relaxation of acetylcholine by about 53% (*p* = 0.01), with a simultaneous minimum effective concentration of 5 × 10^–6^ M, and this value was not significantly different from the value of the maximal relaxation observed in the control rats. The administration of suramin to diabetic animals restored the relaxation properties of acetylcholine to those found in non-diabetic animals (Figure 8C).

## 3. Discussion

In this study, we demonstrated the following actions of suramin in short-term streptozotocin-induced diabetic rats: (a) the decrease of VEGF-A protein expression in the renal cortex and influence on the intraglomerular distribution of VEGF-A protein, consisting of the decreased expression in the central part with the stimulation expression in an area corresponding with the location of glomerular parietal cells; (b) the reduction of increased *Vegfr2* mRNA expression in the renal cortex; (c) an increased decrease in concentration of a soluble form of the vascular adhesion molecule-1 (sVCAM-1) and (d) the restoration of the vasorelaxant properties of acetylcholine to the values found in non-diabetic animals. Moreover, suramin induced a decrease in renal VEGF-A protein expression in non-diabetic rats, suggesting that VEGF-A depletion in the kidney is not dependent on a high glucose concentration.

It is postulated that growth factors, such as VEGF-A, are involved in the pathogenesis of DN. However, the current reports on their actual role are unclear, pointing to their adverse effect, in particular on the permeability of the glomerular filter to albumin and to their beneficial effect on the other [22]. Another factor hindering the analysis of changes in the kidneys in the course of diabetes is the high variability of biochemical and molecular responses, depending on the duration of the disease. An incomplete understanding of the early events in the pathogenesis of DN is one of the main reasons why therapy can slow down the rate of pathology development. Currently, there are no known drugs that could reverse the pathological process developing in the kidneys during the course of diabetes. Therefore, we decided to use a model of short-term streptozotocin-induced diabetes to investigate the role of VEGF-A in the early stages of the disease. Additionally, we used suramin to modify the expression or concentration of VEGF-A, which, as we showed earlier, increases the excretion of VEGF-A in the urine and, at the same time, does not significantly change the concentration of this cytokine in the blood.

Suramin is a broad-spectrum antagonist of membrane purinoreceptor P2, but it is also taken up by endothelial cells and accumulates in the nucleus, suggesting possible action through an intracellular mechanism [37]. It is also used clinically in African trypanosomiasis, cancer, autism and inflammatory diseases of the cardiovascular system [31].

First, we evaluated the VEGF-A protein levels in short-term diabetic glomeruli and demonstrated a 54% decrease in VEGF-A protein levels in glomeruli isolated from 3-week-old diabetic rats. This finding is in accordance with a previous report indicating that, following 1 week of diabetes, VEGF-A levels in rat glomeruli decreased by 38% [38]. Furthermore, exposing mesangial cells to the high glucose concentration medium decreased levels of VEGF-A protein by 50% and mRNA by 27% [39]. It should be noted that there are conflicting reports regarding our results that indicate elevated expressions of VEGF-A in rodent kidneys; however, these are in mice glomeruli [40] or rat renal cortex but not glomeruli [20]. In humans, decreased levels of VEGF-A protein and mRNA have been reported in the glomeruli of patients with diabetes [39,41]. Next, using the western blot analysis, we did not observe a statistically significant effect of suramin on VEGF-A protein levels in glomeruli isolated from non-diabetic or diabetic rats. However, our immunofluorescence data analysis has shown that suramin induces a decrease in the VEGF-A signal in glomeruli isolated from non-diabetic and diabetic rats. Additionally, we found that suramin induces changes in the intraglomerular distribution of the VEGF-A protein. It is clear that the signal from VEGF-A in the central part of the glomerulus is weak but much increased in peripheral areas along the cells lining Bowman’s capsule. Considering that changes in the fluorescence signal correspond to the location of the parietal epithelial cells, the obtained results suggest that suramin may induce phenotypic transitions in parietal epithelial cells. This assumption requires further in-depth research, especially if it is taken into account that the mechanism of parietal epithelial cell transition is rarely understood [42].

The possible biological effects of VEGF-A should not be considered solely on the basis of VEGF-A protein expression. In addition, the level of specific receptors for this cytokine and the duration of diabetes should be taken into account, because there is evidence about increased binding of VEGF to its receptor in the endothelium of mildly injured glomeruli but decreased binding in severely injured glomeruli [28]. For this reason, we measured mRNA expression in the renal cortex for two receptors, *Vegfr1* and *Vegfr2*. We did not observe any significant changes in *Vegfr1* mRNA expression in diabetes or under the action of suramin. However, the diabetic rats had elevated levels of *Vegfr2* and, interestingly, suramin reversed this increase. The effect of short-term diabetes on *Vegfr2* has been described before [20]. However, our results show for the first time the effect of suramin treatment on the decreasing *Vegfr2* expression level. The specific roles of VEGFR-1 and VEGFR-2 in DN pathogenesis are not well understood, although it seems that the activation of VEGFR-2 may impair renal function. Selective stimulation of VEGFR-2 by genetic manipulation has been shown to lead to kidney injury without the aggravation of albuminuria [43]. Among the morphological indicators of renal injury caused by VEGFR-2 stimulation, mesangial expansion, macrophage influx and tubulointerstitial injury were found [43]. Instead, the small-molecule inhibitor SU54106, which blocks VEGFR-1 and VEGFR-2, ameliorates diabetic albuminuria in mice [44]. In light of the results of the previous and our present study, suramin, which reduces *Vegfr2* expression in diabetes, appears to be a promising drug which can be used to reduce kidney damage in patients with diabetes.

Physiologically, endothelial cells are in a quiescent resting state and express low levels of adhesion molecules (ICAM-1 and VCAM-1), produce nitric oxide (NO) and do not interact with circulating leukocytes [45]. However, their dysfunction is characterised by elevated levels of these molecules, resulting in adhesion of leukocytes to endothelial cells during atherosclerosis, hypertension and diabetes [46,47]. The cleavage release of the ectodomain of adhesion molecules from the endothelial cell surface results in the formation of soluble forms, sICAM-1 and sVCAM-1, and the rate of this process is related to the increased expression of membrane proteins [48]. There is evidence that even 24-h exposure of endothelial cells to high glucose concentrations leads to their activation associated with increased expression of VCAM-1 and ICAM-1 [49]. Moreover, VEFG-A stimulates the expression of ICAM-1 and VCAM-1 [12]. In the present study, we did not observe significant changes in sICAM-1 in the blood of diabetic rats, but even the sVCAM-1 concentrations in the blood decreased. These results suggest that systemic endothelial inflammation is unlikely to be elevated, if at all, in short-term streptozotocin-induced diabetes. In contrast, elevated plasma concentrations of sICAM-1 and sVCAM-1 in patients with diabetes have been shown to be associated with the progression of microalbuminuria to macroalbuminuria [50,51]. In our experiments, suramin did not affect the concentrations of sICAM-1 or sVCAM-1 in non-diabetic and diabetic rats, but it should be noted that increased expression of ICAM-1 protein in the diabetic renal cortex has previously been shown and suramin completely blocked the increased expression of ICAM-1. However, sICAM-1 and sVCAM-1 concentrations were not measured in this study [33].

Endothelial cells control the tone of the underlying vascular smooth muscle via the actions of several vasodilators, including NO, prostanoids (e.g., prostacyclin) and endothelium-derived hyperpolarising factor. Their relative contribution varies across different vascular beds, but NO appears to be the dominant player in renal endothelial vasodilator function in non-diabetes conditions. Reduced NO function is a hallmark of microvascular disease and appears to be the dominant factor in the dysfunction of endothelium-dependent vasodilation in diabetes [52,53]. Consistent with other findings [54,55], we found that the renal endothelium-dependent relaxing response to acetylcholine was attenuated in short-term diabetes. The mechanism of diabetes-induced attenuation of acetylcholine action is related to several issues, including reduced NO bioavailability and endothelial nitric oxide synthase (eNOS) uncoupling, leading to the formation of superoxide anions instead of NO [56]. In the present study, suramin restored the endothelium-mediated vasorelaxation of the renal interlobar arteries. The mechanism of action of suramin is currently unknown. Whether the effect is related to the effect of suramin on intracellular metabolism or probably by changing the activity of the VEGF-A/VEGFR axis has not been identified. It has been shown that chronic treatment with VEGF-A preserves acetylcholine-evoked vascular responses in streptozotocin-induced diabetic rats and this effect would be accompanied by normalisation of superoxide anion and NO levels and of endothelial NO synthase expression [57].

In conclusion, the administration of suramin affects the renal VEGF-A/VEGF receptors axis and has a beneficial impact on endothelium-dependent relaxation of renal arteries in short-term streptozotocin-induced diabetes in rats. Therefore, it is reasonable to assume that suramin may be considered a pharmacological agent modifying the activity of the VEGF-A/VEGFR axis, which may be used in research on the potential role of VEGF-A in the pathogenesis of renal vascular complications in the course of short-term diabetes.

## 4. Materials and Methods

### 4.1. Ethical Approval

The experiments were conducted in accordance with the European Convention for the Protection of Vertebrate Animals Used for Experimental and Other Scientific Purposes and approved by the local Bioethics Commission in Bydgoszcz, Poland (approval no. 44/2019).

### 4.2. Animals

The experiments were performed on male Wistar rats (*n* = 28) (Tri-City Academic Laboratory Animal Centre, Gdańsk, Poland), weighing 200–250 g, aged 8–10 weeks at the beginning of the experiments, housed under a 12-h light/12-h dark cycle, and fed a standard pellet diet (Labofeed B, Kcynia, Poland) and water ad libitum. The rats were randomly allocated into four groups (*n* = 7 per group):Non-diabetes group—control group (CON): citrate buffer injected on day –7, saline injected on days 0 and +7.Suramin-treated non-diabetes group (SUR): suramin (10 mg/kg, *ip*) injected on days 0 and +7 after the citrate buffer injection on day –7.Diabetes group (STZ): streptozotocin (60 mg/kg, *ip*) injected on day –7, saline injected on days 0 and +7.Suramin-treated diabetes group (STZ+SUR): streptozotocin (60 mg/kg, *ip*) injected on day –7 and suramin (10 mg/kg, *ip*) injected on days 0 and +7.

The streptozotocin and suramin were injected in volumes of 500 µL. The experiments were performed on rats with tail blood glucose concentrations greater than 11 mM, measured 7 days after the STZ injections. The efficiency of hyperglycaemia induction was 87.5%. Twenty-four-hour urine samples were collected in metabolic cages (Tecniplast, Italy) one day prior to the STZ injection (–8) and on days –1 and +13 after the STZ injection. The urine was collected in tubes containing protease inhibitors (5 × 10^–4^ M PMSF, 10^–6^ M leupeptin) and 3 × 10^–3^ M NaN_3_. At the end of the experiment on day +14, all animals were overdosed with anaesthesia, their thoraxes were opened and blood was drawn by cardiac puncture to preserve the serum of each rat. A schema of the experimental protocol is presented in Figure 9.

### 4.3. Preparation of Renal Interlobar Arteries

The animals, previously anaesthetised with isoflurane inhalation anaesthetic (2.5%, flow rate 0.5 L/min), were killed by an intraperitoneal administration of a lethal dose of pentobarbital (120 mg/kg b.w.). The kidneys were removed immediately and placed in a cold preparation solution (in mM): 146 NaCl, 4.5 KCl, 1.2 NaH_2_PO_4_, 1.0 MgSO_4_, 5.5 glucose, 0.025 EDTA-Na, 5 HEPES, 0.1 CaCl_2_ and pH 7.4. The kidneys were sliced along the main axis of the tubular and vascular structures. The interlobar arteries were dissected, and the arterial sections (length approximately 2 mm) were isolated.

### 4.4. Measurements of Vascular Responses

The interlobar arteries (ILA) were mounted on 40 μm diameter stainless steel wires in a multi-chamber small vessel wire myograph (model 620M, DMT, Denmark) filled with cold preparation solution. The experimental buffers were heated to 37 °C and aerated with carbogen (95% O_2_, 5% CO_2_) for 20 min before the experiments. After mounting, the solution was changed to an experimental solution (in mM): 119 NaCl, 4.7 KCl, 1.2 KH_2_PO_4_, 1.2 MgSO_4_, 5.5 glucose, 25 NaHCO_3_, 2.5 CaCl_2_, 0.03 EDTA-Na and pH 7.4. The myograph was heated up to 37 °C under the unlimited gas flow of carbogen. Thereafter, the vessels underwent a normalisation procedure to an internal circumference equivalent to 90% of that produced under an intramural pressure of 100 mmHg. The viability of the ILA was demonstrated by the application of a high potassium solution (in mM): 123.7 KCl, 1.2 KH_2_PO_4_, 1.2 MgSO_4_, 5.5 glucose, 25 NaHCO_3_, 2.5 CaCl_2_, 0.03 EDTA-Na and pH 7.4. The endothelial-induced vessels relaxation was tested by the cumulative concentration–response relationships to acetylcholine (ACh) (10^–9^ to 10^–5^ M) on arteries precontracted with phenylephrine to 50–70% of the maximum arterial force to high potassium solution, and arterial relaxation was induced on stable phenylephrine-induced precontracted ILA. The data were digitised using the LabChart 8 system (ADinstruments, Dunedin, New Zealand).

### 4.5. Relative Quantitative Real-Time RT-PCR Analysis

The RNA was isolated from the renal cortex using the Chomczynski procedure [58] with our own modifications. The kidney tissues were homogenised in a sterile tube with 1 mL RNA extraction buffer (TRIzol Reagent, Gibco, Grand Island, NY, USA). The extraction was incited by the addition of 250 μL chloroform. After vigorous shaking, each sample was incubated at 4 °C for 15 min and spun down (10,000× *g*) for 15 min at 4 °C. The upper aqueous phase was transferred to a new tube and isopropanol was added in a ratio of 1:2 (isopropanol: RNA Extracol, *v*/*v*). The RNA precipitation was carried out overnight at –20 °C, and the next samples were centrifuged (10,000× *g* for 15 min at 4 °C). The RNA pellet was washed first with 99.8% ethanol and then with 75% ethanol (*v*/*v*), air-dried and reconstituted in nuclease-free water (15–20 μL) and stored at –20 °C. The quantity of the isolated RNA was determined by fluorometry using the Qubit RNA HS assay kit by Qubit^®^ 2.0 Fluorometer (Life Technologies, Grand Island, NY, USA) according to the manufacturer’s instructions. The gene expression levels of *Vegfr1* and *Vegfr2* were determined by real-time RT-PCR performed on a Light Cycler 480 II (Roche Diagnostic GmbH, Mannheim, Germany) using the Path-ID Multiplex One-Step RT-PCR Kit (Thermo Fisher Scientific, Waltham, MA, USA) and the corresponding Universal Probe Library for the rats (Roche Applied Science, Penzberg, Germany). The sequences of primers and probes used for *Vegfr1* and *Vegfr2* are shown in Table 2. The reaction mixture in the final volume 10 μL contained 5 μL of Multiplex RT-PCR Buffer, 1 μL of Multiplex Enzyme Mix and 0.5 μL of each primer for target transcript, 0.2 μL of a target probe, 0.2 μL of primers’ reference gene, 0.2 μL of probe for reference transcript and 2 μL of total RNA. Reverse transcription was performed at 48 °C for 10 min, followed by 95 °C for 10 min. This was followed by 45 cycles of amplification at 95 °C for 10 s and then 60 °C for 40 s. The transcript levels of the target genes were normalised to that of the reference *β-actin* gene *(Actb).*

### 4.6. Western Blotting Analysis

The glomeruli were isolated using the sieving technique and then lysed in buffer (20 mM Tris, 140 mM NaCl, 2 mM EDTA, 10% glycerol and 0.5% Triton X-100) containing Protease Inhibitor Cocktail for 30 min and centrifuged at 13,000× *g* for 20 min at 4 °C. Identical amounts of total protein were denatured (98 °C, 5 min) and then subjected to 10% sodium dodecyl sulphate-polyacrylamide gel electrophoresis. The proteins were transferred to membranes (2 h at 200 mA) and the membranes were blocked (5% BSA, 0.05% Tween-20 in TBS) for 1 h at room temperature. Next, the membranes were incubated at 4 °C overnight with primary antibody (rabbit anti-VEGF-A, 1:500; Merck KGaA, Cat. #AB1876-I; rabbit anti-β-actin, 1:1000, Sigma-Aldrich, A2066). After washing (0.01% Tween-20 in TBS), secondary antibodies conjugated to horseradish peroxidase (anti-rabbit IgG; BD Pharmingen, 554021) were added to the membranes for 1 h at room temperature. The reaction products were detected using SuperSignal West Pico PLUS Chemiluminescent Substrate (Thermo Fisher Scientific, Waltham, MA, USA). The membranes were analysed and archived in a GelDoc-It Imaging System (UVP). The rolling disc method was used to subtract the background signal.

### 4.7. Immunofluorescence Analysis and Confocal Microscopy of Kidney Sections

The kidneys were dissected, fixed in formalin, washed 3 times for 30 min in phosphate buffered saline (PBS) and soaked in 12.5% sucrose for 1.5 h and in 25% sucrose for no less than 24 h. All the incubations were performed at 4 °C. Next, the tissues were washed in PBS, embedded in Tissue-Tek compound, frozen in liquid nitrogen and sectioned into 20 µm slices using a cryostat (Leica). The sections were washed in PBS for 10 min, permeabilised in PBS/0.1% Triton X-100 for 20 min, and then blocked for non-specific antibody binding sites in PBS/3% BSA for 1.5 h. For protein detection, the sections were incubated overnight at room temperature with primary rabbit polyclonal anti-VEGF-A antibody (Merck KGaA, Cat. #AB1876-I) diluted in PBS/3% BSA (1:100). As a negative control, some sections were incubated without the primary antibody. Next, the sections were washed for 5 × 6 min with PBS/0.1% Triton X-100 and incubated for 1.5 h with secondary Cy3-conjugated goat anti-rabbit antibody (Jackson ImmunoResearch, Cambridge House, St. Thomas’ Place, UK) diluted in PBS/3% BSA (1:500) at room temperature. Finally, the sections were washed for 10 min in PBS and mounted in Vectashield medium with DAPI (Vector Labs, Newark, NJ, USA). As a negative control, we used sections prepared without incubating with the primary antibody. Immunofluorescence (IF) analysis was performed on kidneys from five rats from each group, and each staining was repeated three times with a Zeiss LSM 710 confocal microscope (Carl Zeiss, Jena, Germany) using a ×40 objective and Zeiss ZEN software. ImageJ software (NIH, Bethesda, MD, USA) was used to measure the mean fluorescence in each glomerular section. The signal intensity was manually quantified to generate a mean grey value, the sum of the grey values in the selected area divided by the number of pixels within that area.

### 4.8. Analytical Methods

The blood glucose was determined with an Accu-Chek Performa glucometer (Roche, Basel, Switzerland), and the urine volume was determined gravimetrically. Immunoenzymatic assays were used to measure the concentration of rat sVCAM-1 (MyBioSource, San Diego, CA, USA, Cat. No. MBS762680) and rat sICAM-1 (MyBioSource, San Diego, CA, USA, Cat. No. MBS266128). The creatinine (Wiener lab., Rosario, Santa Fe, Argentina) and urea (BioSystems, Barcelona, Spain) concentrations were measured by the enzymatic method.

### 4.9. eGFR Analysis

The GFR was estimated (eGFR, µL/min) based on the serum creatinine <52 µmol/L and urea concentrations according to the published formula [59]:eGFR = 880 × W^0.695^ × C^−0.660^ × U^−0.391^(1)
where W is weight (g), C is creatinine concentration (µmol/L) and U is urea (mmol/L).

### 4.10. Nutrition Variable Analysis

The variables were calculated using the following formulas: feed efficiency ratio = (body weight gain/food intake) × 100; efficiency of food utilisation for body weight = weight gain/daily kilocalories eaten; specific rate of body mass gain = (final body weight—initial body weight)/initial body weight.

### 4.11. Statistical Analysis

Statistical analyses were performed using Statistica 13.3 (TIBCO Software, Palo Alto, CA, USA) and GraphPad Prism 5.0 (GraphPad Software, San Diego, CA, USA). A Shapiro–Wilk test was used to test the determined the normality of the distribution of variables; continuous variables were expressed as mean ±SE (standard error) or median and 25th and 75th percentile. The statistical significance of the differences between the groups was determined with an unpaired *t*-test or Mann–Whitney U-test. A paired *t*-test or Wilcoxon test was used to compare the values of the two related samples. The statistical significance between the groups was determined using two-way or one-way ANOVA and post hoc Tukey’s multiple comparisons. Differences were considered significant for *p* < 0.05.

### 4.12. Materials

The streptozotocin and Triton X-100 were purchased from Merck KGaA (Darmstadt, Germany). The suramin sodium was purchased from Santa Cruz Biotechnology (Dallas, TX, USA, Cat. No. SC-507209). The protease inhibitor cocktail, BSA, acetylcholine chloride, phenylephrine hydrochloride and Tween-20 were purchased from Sigma-Aldrich (Saint Louis, MO, USA). All the other agents were purchased from Avantor Performance Material Poland S.A. (Gliwice, Poland).

## Figures and Tables

**Figure 1 pharmaceuticals-16-00470-f001:**
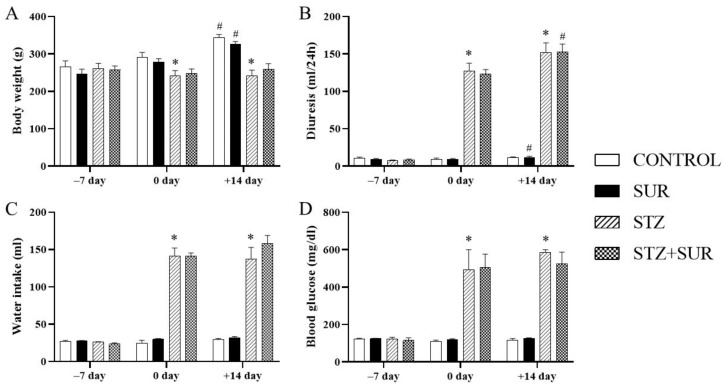
Functional variables output on days –7, 0 and +14 in non-diabetic and diabetic rats treated with suramin: (**A**) body weight, (**B**) diuresis, (**C**) water intake and (**D**) blood glucose. Streptozotocin (60 mg/kg, *ip*, STZ) or citrate buffer (CON) were injected on day –7 and suramin (10 mg/kg, *ip*, SUR) or saline on days 0 and 7. The results are presented as mean ± standard error of the mean or median and 25th–75th percentile, *n* = 7 each group. Statistical significance: ^#^
*p* < 0.05 vs. day 0 in the same group (paired *t*-test/Wilcoxon test), * *p* < 0.05 vs. CON day 0 (unpaired *t*-test/Mann Whitney U-test).

**Figure 2 pharmaceuticals-16-00470-f002:**
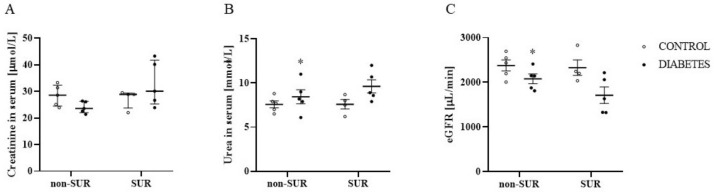
The effects of suramin on creatinine (**A**) and urea (**B**) concentrations in serum and estimated glomerular filtration rate (eGFR) (**C**) in non-diabetic and diabetic rats. Suramin (SUR, 10 mg/kg) or saline (non-SUR) were injected into non-diabetic rats (CONTROL) and streptozotocin-induced diabetic rats (DIABETES) on days 0 and +7 after streptozotocin or citrate buffer injection on day –7. The results obtained on day +14 are presented as single data points with median and 25th–75th percentile (creatinine) or with mean values ± standard error of the mean (urea, eGFR). Statistical significance: * *p* < 0.05 vs. control non-SUR (two-way ANOVA).

**Figure 3 pharmaceuticals-16-00470-f003:**
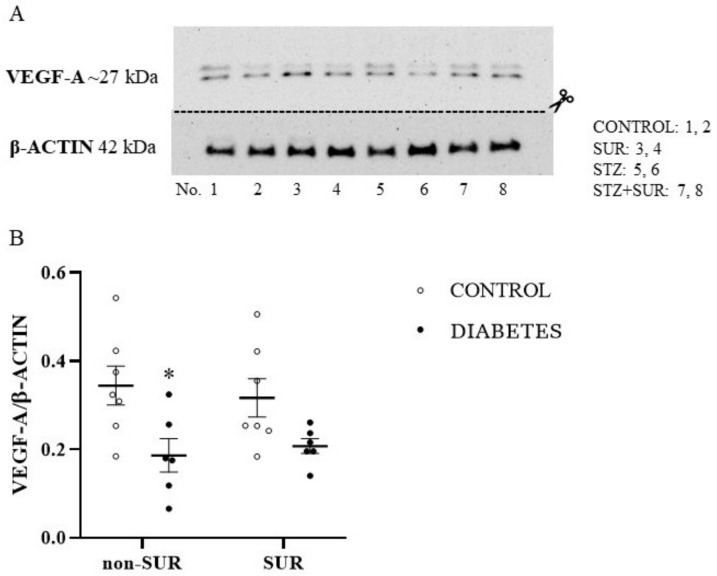
The effects of suramin on VEGF-A protein expression in glomeruli isolated from non-diabetic and diabetic rats. Suramin (SUR, 10 mg/kg) or saline (non-SUR) were injected into non-diabetic rats (CONTROL) and streptozotocin-induced diabetic rats (DIABETES) on days 0 and +7 after streptozotocin or citrate buffer injection on day –7. (**A**) Densitometric scans: relative optical density expressed as the VEGF-A/β-actin ratio. The results of VEGF-A/β-actin expression are presented as single data points with mean values ± standard error of the mean. Statistical significance: * *p* = 0.0023 vs. control non-SUR (two-way ANOVA). (**B**) Representative immunoblot of VEGF-A protein expression in isolated glomeruli; lanes 1, 2: non-diabetic suramin untreated (CONTROL); lanes 3, 4: non-diabetic suramin treated (SUR); lanes 5, 6: diabetic suramin untreated (STZ); lanes 7, 8: diabetic suramin treated (STZ+SUR).

**Figure 4 pharmaceuticals-16-00470-f004:**
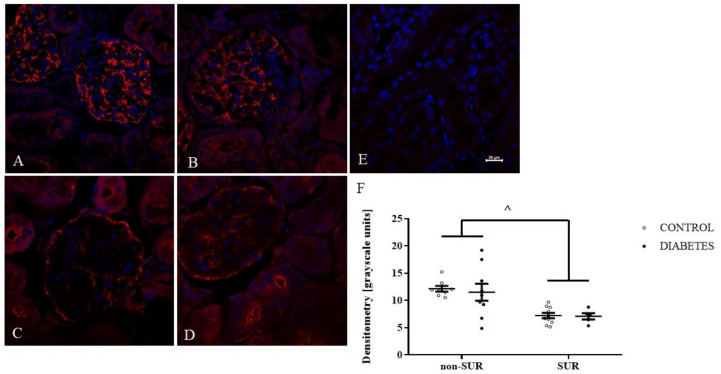
Immunofluorescent localisation and quantitative analysis of vascular endothelial growth factor A (VEGF-A) in glomeruli of non-diabetic and diabetic rats treated with suramin. Suramin (SUR, 10 mg/kg) or saline (non-SUR) were injected into non-diabetic rats (CONTROL) and streptozotocin-induced diabetic rats (DIABETES) on days 0 and +7 after streptozotocin or citrate buffer injection on day –7. Glomerular immunofluorescence staining of VEGF-A (red) and counterstained cell nuclei with DAPI (blue) were analysed by confocal microscopy: (**A**) non-diabetic suramin-untreated; (**B**) diabetic suramin-untreated; (**C**) non-diabetic suramin-treated; (**D**) diabetic suramin-treated; (**E**) kidney sections incubated only with the secondary antibodies (negative control); the bar corresponds to 20 μm; (**F**) quantitative analysis of the VEGF-A fluorescent signal in glomeruli. The mean fluorescence signal associated with VEGF-A in each glomeruli section was measured by ImageJ analysis and quantified manually as a mean grey value; the intensities (mean  ±  standard deviation) are plotted in arbitrary units (a.u). Statistical significance ^ *p* < 0.0001 (two-way ANOVA, Tukey for unequal sample sizes and Bonferroni post-hoc tests).

**Figure 5 pharmaceuticals-16-00470-f005:**
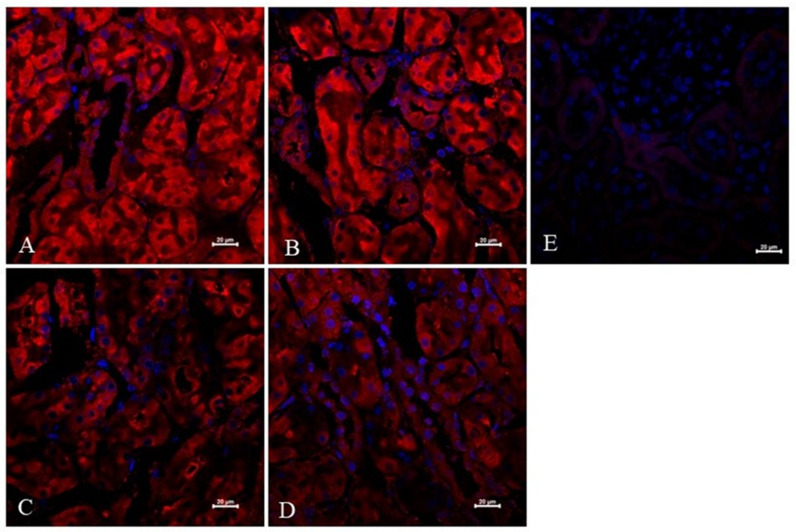
Immunofluorescent expression and localisation analysis of vascular endothelial growth factor A (VEGF-A) in cells of the descending limb of the loop of Henle of non-diabetic and diabetic rats treated with suramin. Suramin (SUR, 10 mg/kg) or saline (non-SUR) were injected into non-diabetic rats (CONTROL) and streptozotocin-induced diabetic rats (DIABETES) on days 0 and +7 after streptozotocin or citrate buffer injection on day –7. Immunofluorescence staining of the descending limb of Henle’s loop for VEGF-A (red) and counterstaining of cell nuclei with DAPI (blue) were analysed by confocal microscopy: (**A**) non-diabetic suramin-untreated; (**B**) diabetic suramin-untreated; (**C**) non-diabetic suramin-treated; (**D**) diabetic suramin-treated; (**E**) kidney sections incubated only with the secondary antibodies (negative control); the bar corresponds to 20 μm.

**Figure 6 pharmaceuticals-16-00470-f006:**
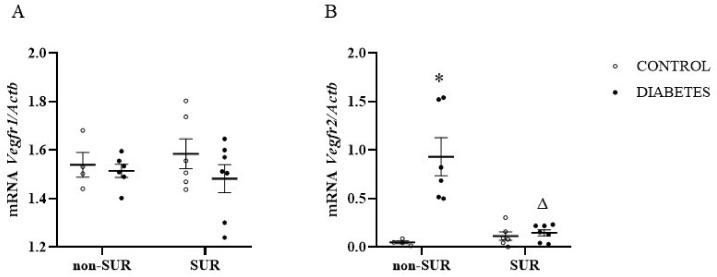
Effects of suramin on *Vegfr1* (**A**) and *Vegfr2* (**B**) mRNA expression in the renal cortex of non-diabetic and diabetic rats. Suramin (SUR, 10 mg/kg) or saline (non-SUR) were injected into non-diabetic rats (CONTROL) and streptozotocin-induced diabetic rats (DIABETES) on days 0 and +7 after streptozotocin or citrate buffer injection on day –7. Quantification of mRNA expression was normalised to the *β-actin gene* (*Actb*) reference transcript. The results are presented as single data points with mean values and standard errors of the mean. Statistical significance: * *p* = 0.0002 vs. control non-SUR, ^Δ^
*p* = 0.0001 vs. diabetic SUR (two-way ANOVA).

**Figure 7 pharmaceuticals-16-00470-f007:**
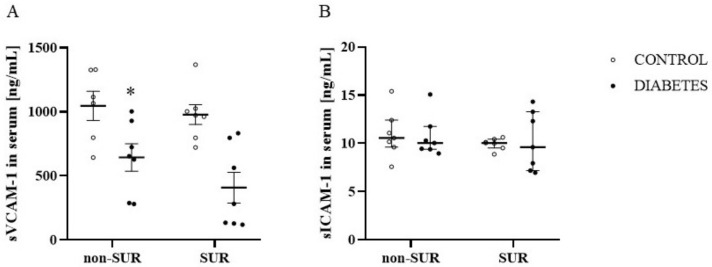
Effects of suramin on blood concentration of soluble vascular cell adhesion molecule-1, sVCAM-1 (**A**) and soluble intercellular adhesion molecule-1, sICAM-1 (**B**) in non-diabetic and diabetic rats. Suramin (SUR, 10 mg/kg) or saline (non-SUR) were injected into non-diabetic rats (CONTROL) and streptozotocin-induced diabetic rats (DIABETES) on days 0 and +7 after streptozotocin or citrate buffer injection on day –7. The results are presented as single data points with mean values ± standard error of the mean (sVCAM-1) or as median and 25th–75th percentile (sICAM-1). Statistical significance: * *p* = 0.0001 vs. control non-SUR (two-way ANOVA).

**Figure 8 pharmaceuticals-16-00470-f008:**
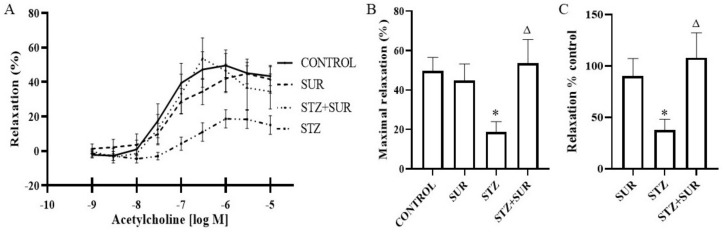
Relaxation effect of acetylcholine on the interlobar artery isolated from non-diabetic and diabetic rats treated with suramin. (**A**) The concentration-dependent effect of acetylcholine on phenylephrine-precontracted interlobar artery, (**B**) the maximal effect of acetylcholine and (**C**) the effect of acetylcholine as the percentage of control set as 100%. Suramin (SUR, 10 mg/kg) or saline (non-SUR) were injected into non-diabetic rats (CONTROL) and streptozotocin-induced diabetic rats (DIABETES) on days 0 and +7 after streptozotocin or citrate buffer injection on day –7. Values are presented as the mean ± standard error of the mean. Statistical significance: B and C * *p* < 0.05 vs. CONTROL, ^Δ^
*p* < 0.05 vs. STZ, two-way ANOVA with Tukey’s post hoc test.

**Figure 9 pharmaceuticals-16-00470-f009:**
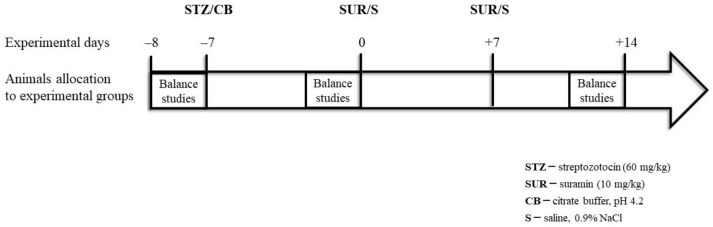
Schema of the experimental protocol.

**Table 1 pharmaceuticals-16-00470-t001:** Nutrition variable output on days +7 and +14 in non-diabetic and diabetic rats treated with suramin. Streptozotocin (60 mg/kg, *ip*, STZ) or citrate buffer (CON) were injected on day –7 and suramin (10 mg/kg, *ip*, SUR) on days 0 and +7.

Parameter	Day	Experimental Groups			
		CON	SUR	STZ	STZ+SUR
Food intake, (g)	0	20 ± 2	23 ± 1	29 ± 1 *	29 ± 1
	+14	23 ± 1	23 ± 0.4	35 ± 2 ^#,^*	35 ± 2 ^#^
Specific rate of body mass gain, (g/kg)	+14	184 ± 31	180 ± 26	−0.51 ± 22.2 ^&^	43 ± 17
Feed efficiency ratio	+14	17 ± 1	15 ± 2	−0.06 ± 1.2 ^&^	2 ± 1
Efficiency of food utilization for body weight, (g/cal)	+14	62 ± 5	56 ± 6	−0.2 ± 4.4 ^&^	9 ± 4

The results are presented as mean ± standard error of the mean or median and 25th–75th percentile, *n* = 7 each group. Statistical significance: ^#^
*p* < 0.05 vs. day 0 in the same group (paired *t*-test/Wilcoxon test), * *p* < 0.05 vs. CON (unpaired *t*-test/Mann–Whitney U-test), ^&^
*p* < 0.0001 vs. CON (unpaired *t*-test/Mann–Whitney U-test).

**Table 2 pharmaceuticals-16-00470-t002:** The sequence of primers and TaqMan probes.

Gene Transcript	Accession No.	Oligonucleotide Sequence 5′-3′	Universal Probe Library Probe
*rFlt1 (Vegfr1)*	D28498	(F) cagtttccaagtggccagag	#22
		(R) aggtcgcgatgaatgcac	
*rFlk1 (Vegfr2)*	U93306	(F) gagacccgcgttttcaga	#65
		(R) aagaacaatatagtctttgccatcc	
*Actb*	Universal ProbeLibrary Rat Actb Gene Assay (Roche, Cat #05046203001)		

## Data Availability

The data presented in this study are available on request from the corresponding author.

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
