# Peer review of "Suramin Affects the Renal VEGF-A/VEGFR Axis in Short-Term Streptozotocin-Induced Diabetes"

_pharmaceuticals, 2023, doi:10.3390/ph16030470_

Round 1

Reviewer 1 Report

General comments:

VEGF-targeting therapy may provide potential benefit against diabetic nephropathy. This study aimed to evaluate the role of VEGF-A in diabetic nephropathy and investigate the effect of suramin which is a broad spectrum anti-angiogenic antagonist, as a pharmacological tool, on the renal VEGF-A/VEGF receptors axis and endothelium-dependent relaxation of renal arteries in a STZ-reduced acute diabetic rat model. As a result, suramin affects the renal VEGF-A/VEGF receptors axis and has a beneficial impact on endothelium-dependent relaxation of renal arteries, and the authors assume that suramin may be useful in research on the potential role of VEGF-A in the pathogenesis of diabetic nephropathy. Despite its limited significance, it provides some interesting insights into the traits and effects of suramin in this short-term diabetic nephropathy model.   

Specific comments:

1. I didn’t see it necessary to use tables (table 1 and 2) presenting these variables. Instead of tables, simple figures would be much appreciated by the audiences.  

2.  Since VEGF-A/VEGFR axis is the main interest of this study, a bit more comprehensive assessments would be needed. For instance, besides the level of mRNA expression (Figure 4), western blot for Vegfr1 and Vegfr2 should be at least evaluated.  

Reviewer 2 Report

1.       A short description of what is known about suramin should also be given in the introduction in addition to those given in the “Discussion”.

2.       Experimental group names are not clear and easy to understand, especially in the figures, although the description of the different groups is well described in the methods section.

3.       It is NOT clear what the authors are comparing in figure 1 and Table 2. Since the authors are investigating the effect of suramin on STZ-induced diabetes, it is imperative that STZ+SUR group should be compared with the STZ group.

4.       Fig. 6. Please check the spelling of “Acetylcholine”. The labeling of the different group is almost understandable in A and B. Does “SUR” mean “Non-diabetic + SUR”? Why are there only 3 groups in figure 6C?

5.       All the figures should be revised to show what exactly the authors have done.

6.       How was the STZ given? IV, IM, IP?

Minor comments

Line 38: “podocyte damage and endothelial cell damage” could be written as “podocyte and endothelial cell damage”

Round 2

Reviewer 1 Report

The authors have responded to the questions point-by-point and revised manuscript properly. I recommend that this original research article should be considered for publication.

Reviewer 2 Report

The authors have addressed all comments